# Fighting Death for Living: Recent Advances in Molecular and Genetic Mechanisms Underlying Maize Lethal Necrosis Disease Resistance

**DOI:** 10.3390/v14122765

**Published:** 2022-12-12

**Authors:** Onyino Johnmark, Stephen Indieka, Gaoqiong Liu, Manje Gowda, L. M. Suresh, Wenli Zhang, Xiquan Gao

**Affiliations:** 1State Key Laboratory for Crop Genetics and Germplasm Enhancement, Nanjing Agricultural University, Nanjing 210095, China; 2China and Kenya Belt and Road Joint Laboratory on Crop Molecular Biology, Nanjing Agricultural University, Nanjing 210095, China; 3Collaborative Innovation Center for Modern Crop Production Co-Sponsored by Province and Ministry, Nanjing Agricultural University, Nanjing 210095, China; 4College of Agriculture, Nanjing Agricultural University, Nanjing 210095, China; 5Biochemistry and Molecular Biology Department, Egerton University, Njoro P.O. Box 536-20115, Kenya; 6Crops Soils and Horticulture Department, Egerton University, Njoro P.O. Box 536-20115, Kenya; 7International Maize and Wheat Improvement Center (CIMMYT), ICRAF Campus, UN Avenue, Gigiri, Nairobi P.O. Box 1041-00621, Kenya

**Keywords:** genomic selection, maize lethal necrosis, marker-assisted selection, resistance mechanism, QTL

## Abstract

Maize Lethal Necrosis (MLN) disease, caused by a synergistic co-infection of maize chlorotic mottle virus (MCMV) and any member of the *Potyviridae* family, was first reported in EasternAfrica (EA) a decade ago. It is one of the most devastating threats to maize production in these regions since it can lead up to 100% crop loss. Conventional counter-measures have yielded some success; however, they are becoming less effective in controlling MLN. In EA, the focus has been on the screening and identification of resistant germplasm, dissecting genetic and the molecular basis of the disease resistance, as well as employing modern breeding technologies to develop novel varieties with improved resistance. CIMMYT and scientists from NARS partner organizations have made tremendous progresses in the screening and identification of the MLN-resistant germplasm. Quantitative trait loci mapping and genome-wide association studies using diverse, yet large, populations and lines were conducted. These remarkable efforts have yielded notable outcomes, such as the successful identification of elite resistant donor lines KS23-5 and KS23-6 and their use in breeding, as well as the identification of multiple MLN-tolerance promising loci clustering on Chr 3 and Chr 6. Furthermore, with marker-assisted selection and genomic selection, the above-identified germplasms and loci have been incorporated into elite maize lines in a maize breeding program, thus generating novel varieties with improved MLN resistance levels. However, the underlying molecular mechanisms for MLN resistance require further elucidation. Due to third generation sequencing technologies as well functional genomics tools such as genome-editing and DH technology, it is expected that the breeding time for MLN resistance in farmer-preferred maize varieties in EA will be efficient and shortened.

## 1. Introduction

Maize is among top three most important cereal crops grown worldwide, with the USA and China being the largest producers of it. However, in the Eastern and Southern Africa (ESA) region, maize is a crucial staple crop providing food, income and jobs within its value chain [1]. Its production is often severely constrained by biotic and abiotic stress factors, among which, the diseases caused by diverse phytopathogens are particularly a serious concern for the farmers and producers. Nonetheless, the emergence of Maize Lethal Necrosis (MLN) disease first reported in 2011 in Kenya is one of the single most important biotic threat to maize production within the ESA region, and it can lead up to 100% crop loss [2,3,4]. Yet, MLN is still a major threat to maize crop health in Eastern Africa and worldwide. Government and non-governmental organizations have developed mitigation measures to curb the MLN spread within the ESA region; however, its spread to other regions inSub-Saharan Africa (SSA) is still looming [5,6,7]. The severity of MLN spreading in the new SSA regions is exacerbated by lack of efficient control measures, and a few resistant germplasms have been identified thus far (https://mln.cimmyt.org) (accessed on 12 September 2022).

Over the past decade, tremendous efforts have been made towards understanding the occurrence of MLN, its threat to maize production, the causative agents, the identification of a resistant/tolerant germplasm and the surveillance and curtailing of its spread [4,6,7,8]. In addition, functional genomics tools have been used to dissect the molecular and genetic basis of MLN resistance/tolerance, as well as the discovery of key loci associated with resistance/tolerance [7,9]. In this review, we summarize the recent advances in MLN disease resistance, focusing on the molecular and genetic approaches for resistance identification, QTL mapping and genome-wide association studies (GWAS) of MLN disease resistance loci. Furthermore, the potentials of using molecular genomics tools for the improvement of MLN resistance and the development of resistant lines via marker assisted breeding (MAS), genomics selection (GS) and genome editing are also discussed.

## 2. Occurrence and Threats of MLN Disease in SSA

While the major type of MLN-causing virus maize chlorotic mottle virus (MCMV) was first described as early as 1974 in Peru, MLN was first reported in Longisa Division Bomet County, Kenya in September 2011 [10] and in Yunnan, China [10]. The symptoms resembling those of MLN were observed by the year 2012 in multiple counties within the former Central, Nyanza, Western and Rift Valley provinces of Kenya [10]. The disease rapidly emerged in multiple regions within EA, including Rwanda [11,12] and Democratic Republic of Congo (DRC) [13], Uganda [14], Tanzania [10], South Sudan and Ethiopia [3] (Figure 1). The rapid spread of the disease within EA was exacerbated by the fact that more than 95% of the commercial maize lines in these regions are susceptible to MLN [3]. MLN disease usually displays mottling symptoms on the leaves, which can start from the base of young leaves in the whorl and extend to other tissues, stunting and shortened the internodes, causing dead heart and necrosis, as well as sterility, poor seed setting and shriveled seeds (Figure 2). The disease causes irreversible damage and the eventual senescence of the maize plants before they mature and yield grain. For instance, in the year 2012, Kenya experienced 30–100% crop loss depending on the time of MLN onset and its severity. The loss was estimated to be 0.5 million tons of maize, which is worth USD 180 million [10,15]. The highest losses were reported in the highlands of central Kenya and Rift Valley in 2013 [16]. Moreover, in the year 2018 it was estimated that small-scale farmers in Eastern Africa alone lost approximately USD 291–339 million due to MLN [4]. Due its rapid spread, impact on livelihoods and economies of SSA, there is need for robust mitigation measures and a concerted response to MLN from multiple stakeholders within the region and beyond it.

## 3. MLN Causative Agents and Disease Management

Numerous lines of evidence point out that MLN is often caused by the synergistic co-infection of MCMV and any member of the *Portyviridae* family [4,17,18]. Synergistic interactions of potyviruses and other viruses have been reported, and in such a scenario, non-potyviruses are the major beneficiaries [19,20]. The increased accumulation of MCMV in co-infected plants is attributed to the presence of potyviruses which bring about unilateral synergism [19,21]. It has been hypothesized that an increased concentration of MCMV in the co-infected plants in comparison to those infected by MCMV alone is due to suppressed the host response by the potyvirus [19].

### 3.1. Maize Chlorotic Mottle Virus (MCMV)

MCMV is the monotypic species of the genus Machlomovirus in the family *Tombusviridae* [22]. The virus was first reported in Peru [23], and again later in the United States of America [18]. MCMV has also been reported in China [24], Kenya [10], Uganda [3,6] and Rwanda [11]. Maize is among the natural hosts of MCMV, and its infection symptoms range from mild to severe, and they are characterized by leaf necrosis, chlorotic mottling, malformed or partially filled ears, stunted growth and eventual plant death [25].

MCMV has a 4.4 Kb single-stranded positive RNA genome that structurally resembles panicovirus, albeit, with an additional open reading frame (ORF) at the 5′ end [26]. MCMV has seven strategically expressed proteins that are encoded by five ORFs. The first ORF (ORF1) encodes p32 protein, whilep50 and its readthrough protein are encoded by ORF2 which overlaps with ORF1 [27,28]. In addition, the p7a, p7b, p31 and CP proteins are translated from the sub-genomic RNA1 (sgRNA1), spanning nucleotides 2971–4437 on the genomic RNA [27,28,29]. A mutagenesis analysis has demonstrated that the MCMV movement from cell to cell is facilitated by the expression of p7a, p7b and CP [28]. The p31 protein is expressed as a readthrough extension of p7a, and it is a major pathogenicity determinant of MCMV. Furthermore, the expression of p31 by the virus vectors in the systemically infected leaves induced necrosis [30]. The 337 nucleotide long noncoding sgRNA2 has been shown to accumulate in MCMV-infected maize protoplasts and plants [31].

Thrips, corn rootworms, Chrysomelid beetles including *Diabrotica* spp. [32] and the corn flea beetle [33] are considered to be MCMV vectors. The early study showed that both the larval and adult forms of *Oulema melanopa* are able to transmit MCMV, and the transmission rate for the larvae and adults are 0.905 and 0.390, respectively [34]. *Frankiniella williamsi* (maize thrips) has also been identified as the major vector in East Africa [3,10,33,34]. Using a leaf disk transmission assay and a real-time RT-PCR to examine MCMV transmission by corn strips, Cabanas et al. found that both the larvae and adults of *F. williamsi* were capable of transmitting the MCMV virus for up to six days after acquisition, whereas the transmission rates decreased as the time progressed [33]. However, MCMV can also be transmitted via maize seeds [35,36]. Earlier experiments have shown that the rates of MCMV transmission to progeny plants range from 0–0.33% [17,35]. MCMV has also been reported to induce changes in the host plants’ volatiles which attract vectors such as thrips [37].

### 3.2. Potyviruses

Potyviruses are among the most widely distributed group of plant viruses with economic significance, and they account for 30% of the viruses which infect plants [38]. The economically important potyviruses includes sugarcane mosaic Virus (SCMV) [39], maize dwarf mosaic virus (MDMV) [40,41], Johnson grass mosaic virus (JGMV) [42] and sorghum mosaic virus (SrMV) [43]. In Kenya, for example, SCMV was found in 15.2%, 15.8% and 19.6% of the sampled maize fields in the Nyanza, Rift valley and Western Kenya regions, respectively [44]. The genome of the potyviruses possess a single ORF, and when this is translated, it yields a poly-protein that is cleaved into ten mature proteins namely; P1 protease, helper component protease (HC-Pro), P3, 6K1, cylindrical inclusion (CI) protein, 6K2, viral genome-linked protein (VPg), nuclear inclusion a-protease (NIa-Pro), nuclear inclusion b (NIb), CP and three viral proteases [45]. In addition, ribosomal frame-shift or polymerase slippage generates the P3N-PIPO protein, which facilitates virus cell-to-cell movement [46,47]. Potyviral HC-Pro is well characterized, and it targets various steps of RNA silencing to block the plants’ defense response [48,49]. HC-Pro also aids virus transmission by aphids [50], the production of viral particles [48], symptom development [51], genome replication [51] and virus movement [52,53].

Aphid species, such as *Myzus persicae*, *Rhopalosiphum padi* L., *Schizaphis graminum* and *Rhopalosi phummaidis,* among others, are reported to aid the transmission of potyviruses worldwide. They tend to be ubiquitous where maize grows, such as in the ECA region. Among the reported alphid vectors, *M. persicae* seems to be the most efficient and/or widely studied vector, as it is capable of transmitting 53.4% of the 176 potyviruses [53]. Aphid probing and feeding behaviors have largely been responsible for the quick acquisition and transmission of portyviruses, which occurs within minutes. For instance, as a sedentary species, but as an excellent vector of PVY, *M. persicae* showed a reduced non-probing duration but increased phloem sap ingestion and arrestment on the host plants infected by PYV compared to those of *M. euphorbiae*, a mobile species that is not an ideal vector of PVY [53].

### 3.3. Other MLN Associated Viruses

In addition to MCMV and SCMV, other infecting viruses such as maize streak virus (MSV) which belongs to the family *Geminiviridae* have also been found to co-infect maize plants with MCMV, leading to MLN disease [3]. Furthermore, Next Generation Sequencing (NGS) has revealed previously unknown viruses that co-occur with MCMV, such as maize yellow dwarf virus (MYDV)-RMV, maize yellow mosaic virus (MYMV) and barley virus G (BVG) belonging to the *Polerovirus* genus, causing MLN in Kenya [54]. The presence of *Polerovirus* in sugarcane has also been described in Brazil and Nigeria, and their partial genome sequences in maize have been reported in the USA [55]. MYMV (*Polerovirus* genus) was also detected in maize and sorghum by RT-PCR in Kenya, which was believed to co-infect with MCMV and SCMV, leading to MLN disease [56].

## 4. MLN Phenotyping and Management

### 4.1. Phenotyping and Diagnosis of MLN Disease

Phenotyping in East Africa is majorly conducted jointly by the International Maize and Wheat Improvement Center (CIMMYT) and the Kenya Agricultural & Livestock Research Organization (KALRO) in Kenya [57]. A screening facility for MLN is located in Naivasha, Kenya (https://mln.cimmyt.org) (accessed on 12 September 2022). Several robust, rapid and sensitive diagnostic approaches have been developed in Eastern Africa for both MCMV and SCMV to enable disease surveillance and preventive measures. Currently, the diagnosis of individual MLN viruses heavily relies on an Enzyme Linked Immunosorbent Assay (ELISA), Reverse transcriptase PCR (RT-PCR) and NGS [3,10,58]. The RT-PCR and NGS methods are more sensitive compared to ELISA since they target the identification of a specific strain or species of MLN virus partners [54]. For ELISA, commercial kits employing the Double Antibody Sandwich (DAS) for detecting SCMV and MCMV are available. Furthermore, RT-PCR and quantitative RT-PCR (qRT-PCR) assays have been developed for SCMV. However, assays for the simultaneous detection of all of the known virus isolates have not been tested [59]. The NGS approach is expensive for routine analyses, but it is an excellent tool for characterizing the potyvirus populations [54,59].

The field-based diagnosis of the leaf samples is achieved by use of simple, reliable and cost-effective immune strips. The CIMMYT and National Plant Protection Organizations (NPPOs) in ESA have set protocols for immunostrip-based MCMV diagnosis both in the farmers’ fields and maize production fields [60]. Current efforts are geared towards developing much more robust assays specifically for the viral partners causing MLN, especially SCMV. The major focus is on developing antisera and primers that are specific for the East African virus isolates.

### 4.2. MLN Disease Management

To control MLN disease effectively, several management procedures have been proposed. These included field sanitary measures such as crop rotation to curtail the virus population, disease surveillance, the control of vectors using appropriate pesticides and the use of MLN-resistant maize varieties, among others [7,30,61]. The use of MLN-tolerant/resistant maize lines and hybrids, rapid and robust diagnostics kits for viral partners and surveillance across Africa have been employed [2,7,60]. However, additional measures are inevitable in order to mitigate its spread further. The extra measures entail the production and exchange of MLN pathogen-free maize seeds and regional policies, ensuring that the farmers and partners are informed, and promoting community phytosanitary practices (https://mln.cimmyt.org) (accessed on 12 September 2022). There are notable constrains for some of the suggested measures, for example, practicing crop rotation is a big challenge for smallholder farmers in ESA who inevitably practice continuous maize planting and intercropping [61,62]. In addition, some of the approaches that have worked in countries such as the USA [58] may not be applicable in ESA. Therefore, the most viable and sustainable approach is the development of farmer-preferred resistant maize varieties since linked markers for resistance to SCMV and MCMV are available [63,64].

## 5. Genetic Dissection of MLN Resistance

The genetics of MLN is both a simple and complex combination as it is controlled by the combination of genetic resistance to SCMV, MCMV as well as their interactions. The earlier studies clearly showed that SCMV is controlled by two major-effect genomic regions on chromosome 3 and 6 and many small-effect QTLs distributed on different chromosomes. On the other hand, MCMV had a major-effect resistance gene on chromosome 6 [65,66] and many minor-effect QTLs distributed on chromosome 3, 4, 5 and 8 [67,68]. The genetic studies also support that the SCMV and MCMV and MLN resistance are governed mainly by additive effects, and to some extent by non-additive effects [69,70]. This suggests MLN resistance is controlled by a few major-effect genes and many minor-effect QTLs, and it is possible to improve it by rapid recurrent selection [71]. Moreover, tremendous efforts have gone towards resistance germplasm screening, QTL mapping and GWAS analysis to identify MLN resistance loci or candidate genes. These remarkable efforts led to the successful development of elite resistant donor lines, such as KS23-5 and KS23-6, and the identification of multiple resistance loci on Chr 3 and Chr 6 [65]. Despite this progress, the genetic basis underlying MLN resistance remains poorly elucidated.

### 5.1. QTL Mapping of MLN Resistance

The accumulating evidence suggests that MLN resistance is a complex trait controlled by numerous genetic loci with minor effects [67]. Therefore, in order to develop maize lines that are resistant to existing and emerging MLN viruses, the identification of the genomic loci associated with MLN resistance is necessary [64]. So far, a number of QTLs responsible for potyviruses and MCMV resistance have been mapped in resistant maize genotypes [65,71,72,73]. For instance, some closely linked loci on maize chromosome 3, 6 and 10 conferring resistance to SCMV, MDMV, JGMV, SrMV, WSMV and ZeMV have been reported [63,64,65,72,73,74,75,76,77,78]. These loci were identified in the US cornbelt, European, Chinese and tropical maize germplasms (Table 1). An earlier study successfully mapped three QTLs, two of the QTLs (*Scm2a* and *Scm2b*) on Chr 3, bin 3.04, and one of them (*Scm1*) on Chr 6, bin 6.01 were associated with SCMV resistance in an F2:3 population consisting of 150 lines generated by crossing two tropical maize inbred lines, L520 (resistant) and L19 (susceptible). These QTLs accounted 13.3%, 41.9% and 7.7% of the phenotypic variation explained (PVE) for SCMV resistance, respectively [79]. In another study, a total of 118 QTLs for MLN-disease severity (MLN-DS) and area under disease progression curve (AUDPC) were identified using seven F3 bi-parental populations and 500 KASP markers [72]. Among the 118 QTL identified, seven of them were stable major QTLs located on Chr 3, 5, 6, 8 and 9, spread across seven populations, explaining 10.54~44.50% of the PVE [72]. Although most of the QTLs detected were small-effect loci, qMLN6_85 had AUDPC values explaining 21.6% of the PVE in population 1, and qMLN9_142 detected in population 2 explained 12.72% and 10.54% of the PVE for MLN-DS and AUDPC, respectively. Interestingly, qMLN3_142 explained 28.84% and 11.09% of the PVE in population 3 and 24.44% and 27.46% of it in population 5 for MLN-DS and AUDPC, respectively. Moreover, qMLN3_130 explaining 44.51% and 26% of it for MLN_DS and AUDPC values, respectively, was detected in population 6 [72]. Furthermore, a major-effect QTL (qMLN06_157) that could explain 55–70% of the PVE was mapped on chromosome 6 by linkage mapping using an F2 bi-parental population constructed by KS23-5 and KS23-6 lines, together with GWAS [65] (Table 1). Similarly, through joint linkage mapping combined with GWAS and genomic prediction, SCMV and MCMV resistance loci were identified in three DH populations and one association population consisting of 380 lines. Intriguingly, two major QTLs, qMCMV3-108/qMLN3-108 and qMCMV6-17/qMLN6-17, were found to be localized on Chr 3 and Chr 6 [67], respectively. The reports reviewed, among others, clearly indicate that MLN resistance in maize genotypes indeed has many loci. Therefore, the development of resistant maize lines to existing and emerging MLN viruses necessitates the identification of the key loci associated with MLN resistance [62].

In comparison to SCMV and MLN resistance, the genetic basis for MCMV resistance alone in maize remains largely unexplored [64]. In a previous work, five resistant inbred lines, namely, KS23-6, KS23-5, N211, DR and Oh1V1 were crossed with Oh28, the highly susceptible line, to construct F2 populations and an RIL population (Oh1V1×Oh28), respectively. The populations were then subjected to MCMV disease phenotyping and QTL mapping, which led to the identification of multiple QTLs on Chr 1, 2, 3, 5 and 6. These QTLs explained 9~78% of the PVE [66]. Majority of the QTLs identified were consistent with a separate study conducted using F2 and RIL populations constructed from Oh1V1×Oh28. The QTLs located on Chr 2, 3, 6 and 10 were responsible for the resistance to diverse virus species, including MCDV, SCMV, WSMV, MDMV, MFSV and MMV [64]. In addition, 24 SNPs on eight chromosomes were also linked to MLN disease resistance. In another study, the QTL mapping of three bi-parental populations also revealed a major QTL for MLN resistance on chromosome 3 and 6 with minor QTLs being distributed across nine chromosomes [87]. These findings, together with other reports in the literature strongly suggest the existence of a highly conserved genetic region on chromosomes 3 and 6 for SCMV resistance.

### 5.2. Genome Wide Association Studies (GWAS)

In addition to QTL mapping, GWAS as another strategy has been also utilized to understand the genetic architecture of MLN resistance. It has been used widely particularly to resolve small-effect QTLs associated with MLN resistance. The major strength of the GWAS is its ability to allow a rapid analysis of the genetic architecture with complex traits without need for constructing a linkage mapping population, which is a time-consuming and labor-intensive process [88]. GWAS utilizes linkage disequilibrium analysis to achieve association mapping, which increases the resolution and identity of the preferred genetic loci associated with the trait(s) of interest [89]. GWAS and genomic prediction (GP) studies have been proven to be effective in exploring the complexity nature of MLN resistance [90]. GWAS and GP have revealed that the minor genes for MLN disease traits in maize are distributed across the genome, thus explaining the complex nature of MLN disease resistance [91]. Intriguingly, the QTL Scmv1 on Chr 6 and Scmv2 on Chr 3 that was identified using the linkage-mapping approaches were further verified by GWAS in an association mapping population consisting of 578 lines from the USA [92]. A significant SNP was identified to be localized 5 Mb downstream of the Scmv1 locus, and a strong loci-resistance trait correlation was found between this region and the presence of the allele of the PAV encompassing Scmv1. Furthermore, Scmv2 exists in the lines carrying the Scmv1 locus, and it is highly associated with the resistance trait in these lines [92], further strengthening the importance of Scmv1 and Scmv2 in MLN resistance.

Several previous GWAS works for MLN resistance association conducted by CIMMYT have utilized two major panels, the Drought tolerant Maize of Africa Association Panel (DTMA-AP) and the Improved Maize for Africa Soils Association Panel (IMAS-AP), which consist of 235 and 380 lines, respectively [80,93]. These two populations were constructed mainly from tropical and subtropical maize germplasms [88]. Using high-density markers, GWAS revealed marker trait association for both MLN-DS and MLN-AUDPC, with six and eighteen significant marker–trait associations for MLN resistance identified in the DTMA-AP and the IMAS-AP explaining 37% and 30% of the PVE, respectively [80].

In a most recent study, a GWAS conducted using a maize association panel consisting of 282 lines identified multiple genetic variations associated with MLN resistance on chromosome 3, 4, 5 and 8 [68]. Furthermore, a QTL on chromosome 10 was found by linkage mapping using an RIL population constructed using the susceptible line B73 and the resistant line CML333, which were identified through screening the natural population [68]. On the other hand, GWASs along with genomic prediction (GP) were conducted using a large population consisting of 1400 tropical lines, leading to the identification of 32 SNPs that were significantly associated with MLN resistance, and a number of candidate genes were also proposed [91]. Similarly, deploying a GWAS and GP in another recent study using a DH population revealed multiple significant SNPs for MLN resistance and agronomic traits [81]. Interestingly, many of the significant SNPs identified were for diverse agronomic traits, such as grain-related traits and leaf disease resistance, while 12 and 10 SNPs were determined for MLN-DS and AUDPC, respectively [81].

### 5.3. Cloning of Candidate Genesand Molecular Mechanisms Underlying MLN Resistance

Due to genetic complexity and successful cloning of only a limited number of candidate genes, the molecular mechanisms underlying MLN resistance still remain largely unexplored. Through deploying genetics approaches, several, but still a limited number of MLN resistance genes have been cloned (Table 2). Eukaryotic translation initiation factor 4E (eIF4E) has been identified as a susceptibility factor in many potyviral-plant interactions [94,95]. eIF4E is also usurped by MCMV to bind the viral 3′ cap dependent translation element [29]. Similarly, maize Elongin C protein ZmElc was found to interact with MCMV viral genome-linked protein (VPg) using the Yeast two-hybrid (Y2H) method. The silencing of ZmElc resulted in decreased transcripts of SCMV and ZmeIF4E, but increased levels of MCMV, suggesting a contrasting role of ZmElc in facilitating SCMV and suppressing MCMV virulence [94]. It is envisaged that the elucidation of the roles of the host genes such as ZmeIF4E and ZmElc will open up new opportunities to develop new maize lines to complement the ongoing breeding of MLN-tolerant/resistance maize varieties. In addition, protein disulfide isomerase (PDI) was also reported to support both MCMV and SCMV accumulation [96,97]. However, the roles of the host factors for resistance to MCMV and/or SCMV are not always the same [94]. Remarkably, two major genes, Scmv1 on Chr 6 and Scmv2 on Chr 3, have been identified for SCMV resistance [83,84,85]. Scmv1 encodes a typical thioredoxin-h protein, and it is highly expressed in resistant lines, and it is responsible for a decreased viral RNA accumulation in the tolerant lines [82,86]. Scmv2, which encodes an auxin binding protein 1 (ZmABP1), is located on Chr 3, and it is highly expressed in the resistant lines and acts synergistically with Scmv1 to enhance the virus resistance [84,86,98,99]. However, the greatest challenge is the availability of scanty information on the mechanisms by which these two genes confer resistance to SCMV and other potyviruses. In Eastern Africa, the presence of multiple and complex isolates of SCMV associated with high virulence on the host have been confirmed [93,100]. Therefore, it is prudent to understand how different sources of resistance interact with a range of potyviruses isolates.

In recent years, omics approaches have increasingly been used to validate the candidate genes for SCMV resistance and the elucidation of the corresponding molecular mechanisms. For example, microarray and qRT-PCR have been used to investigate the expression patterns of the genes encoding the metallothionein-like protein, S-adenosylmethionine synthetase, germin-like protein or 26S ribosomal RNA with regard to SCMV and MDMV resistance in four (F7^SS/SS^, F7^RR/RR^, F7^SS/RR^ and F7^RR/SS^) RIL lines [101] (Table 2). Furthermore, systemic proteomics has been employed to identify the novel and differentially expressed proteins associated with SCMV resistance in a pair of lines with contrasting SCMV resistance levels [95,101]. Proteomics has also been used to investigate the mechanism of MCMV resistance in maize [102]. For instance, the iTRAQ-based comparative proteomics approach led to the identification of a series of differentially expressed proteins in B73 line upon MCMV infection. Furthermore, the functional validation of the iTRAQ-identified candidate genes using Cucumber Mosaic Virus-based virus-induced gene silencing (CMV-VIGS) demonstrated that two candidate genes, encoding disulfide isomerases such as protein ZmPDIL-1 and peroxiredoxin family protein ZmPrx5, likely play roles in the hosts’ susceptibility to MCMV infection [96] (Table 2). Overall microarray, qRT-PCR and proteomic data provide potential gene targets for manipulating SCMV resistance in maize.

Although some tolerant maize lines have been developed and are being grown in Hawaii, little is known about the mechanisms by which MCMV resistance occurs [71]. The accumulating evidence suggest that the plant hormones also play essential role in MLN disease resistance. For instance, an increase in the expression of salicylic acid (SA)-responsive pathogenesis-related (PR) protein genes and an overall SA increase was observed in MCMV-infected maize lines [30]. The resistance against MCMV/SCMV by maize seedlings was enhanced when they were exogenously treated with SA [30,103,104]. Moreover, SCMV symptoms were exacerbated when the maize *phenylalanine ammonia lyase* (*ZmPAL*) genes were knocked down [104]. Steroid phytohormones such as Brassinosteroids (BRs) are also involved in the defense response to MCMV infection [105]. Based on the transcriptome sequencing data analyses, the BR-associated genes were up-regulated upon MCMV infection [105]. Unlike the roles of SA, the up-regulation of BRs enhances the susceptibility of the maize seedlings to MCMV infection. The enhanced susceptibility of maize plants with a knocked-down *DWARF4* gene and nitrate reductase gene *ZmNR* to MLN clearly suggests that the BR pathway is involved in maize’s susceptibility to MCMV accumulation in a Nitric Oxide (NO)-dependent manner [105]. However, the regulatory mechanisms underlying the roles of the BRs in plant antiviral immunity require further investigation [8].

Gene expression is tightly regulated by epigenetic modifications; however, maize’s epigenetic mechanism of MLN resistance is not well studied. On the other hand, a recent study revealed that microRNA is likely to be involved in controlling MLN resistance [98]. This study indicated that microR167 (Zma-miR167) produced in maize could modulate MCMV resistance through the direct targeting and cleaving of Auxin Response Factor3 (ZmARF3) and ZmARF30. These two proteins are essential transcription factors of the auxin signaling pathway binding to the promoter of the *Polyamine Oxidase 1* (*ZmPAO1*) gene [98] (Table 2). Reversely, *ZmPAO1* is hijacked by the MCMV p31 protein to facilitate virus growth, thus it functions as a negative regulator of MCMV resistance to encounter the hosts’ defense [98]. In a separate study, the MCMV p31 protein was also found to hijack another host protein maize catalases (ZmCATs) to suppress SA-mediated resistance [30], suggesting that MCMV has evolved a conserved mechanism to interfere with the hosts’ defense system. Interestingly, using four miRNA target prediction algorithms, 10 out of 321 maize-derived miRNAs were found to be capable of silencing the MCMV genome, thereby providing a potential strategy for suppressing MCMV virulence and enhance maize resistance to MLN [106].

## 6. Germplasm Identification and Molecular Breeding for MLN Resistance

Over the past decades, crop breeding strategies have been significantly improved and accelerated by benefiting from the advances and application of novel technologies. The notable advances and technologies include functional genomics, phenomics, doubled haploids, genomic selection, genome editing and synthetic biology, as well as big data and artificial intelligence. These new technologies, when they are combined with the conventional breeding techniques, can enormously facilitate development of maize that is resistant/tolerant to MLN disease.

### 6.1. Identification of Resistance Sources and Marker-Assisted Breeding

In order to combat MLN disease, thus far, the considerable efforts have mainly focused on accession screening to identify the resistant/tolerant germplasms. A conducive environment for MLN-virus buildups has been facilitated by continuous cropping and the use of highly susceptible commercial lines. Therefore, through fast-tracked screening, an understanding of the genetic architecture of MLN disease resistance and development of resistant germplasm is inevitable. Previously, the artificial screening for MLN-resistant germplasms has been conducted in the field and under growth-chamber controlled conditions through direct artificial inoculation in Kenya and Ohio, USA, respectively [3,80,107]. The advantage of artificial inoculation, unlike natural infection, is the efficiency of producing a sufficient amount of viral inoculum, the high transmission rates, a uniform infection, and the clear separation of the resistant and susceptible plants [63,64,71,108]. Due to its high rate of transmission and minimal number of disease escapes, artificial inoculation has been the preferred approach for genetic studies [71].

Both the public and private sectors are consolidating efforts geared towards developing improved germplasms [80,87]. The initial screening of common open-pollinated varieties and commercial hybrids in Kenya revealed high levels of susceptibility under artificial inoculation [87,109]. Thus far, in Kenya, over 120,000 maize accessions, representing worldwide germplasms, have been screened, but only a few tolerant/resistant hybrids have been identified (https://mln.cimmyt.org) (accessed on 12 September 2022). On the other hand, the MCMV screening of 107 tropical maize inbred lines identified 47 lines that were tolerant. The number of the most notable lines was five, namely, KS23-6, KS23-5, N211, DR and Oh1V1, which developed significantly fewer symptoms compared to that of the controls [66]. Moreover, the further screening of an additional 547 lines from the maize diversity and Germplasm Enhancement of Maize (GEM) panels did not reveal lines that developed fewer symptoms than these five lines did [110]. In the germplasm screening trail conducted in Ethiopia, however, only 2% of the three hundred and six lines screened showed a resistance to MLN [36], while in Zambia, only two varieties, GV662A and IICZ3085, were moderately resistant from a population of four hundred and seventy-three lines tested [68,111]. These results obtained from the germplasm screening conducted in the three ESA countries (Kenya, Ethiopia and Zambia) clearly demonstrate that MCMV tolerance is rare, and thus, there is a need to seek alternative sources of resistance.

The first essential step among the effective and important practical solutions for a breeding program is screening the elite maize varieties with an enhanced resistance to MLN. Over the past years, numerous elite and exotic maize sources have been accessed and examined globally for MLN resistance. Specifically, parent inbred lines and corresponding progenies from European pools, including D21, D32 and FAP1360A, have been tested for SCMV resistance [109,112,113]. Four inbred lines (CLRCY039, DTPYC9-F46-1-2-1-2, CLRCY034 and CLWN270) with a high resistance level to MLN were successfully identified out of four hundred and thirty-one tropical elite and low-nitrogen-tolerant maize lines [114]. The other potential sources included the Chinese maize line Hunazao4 [9] and the newly introgressed tropical maize lines [72]. Overall, CIMMYT had conducted a series of screening on more than 120,000 germplasms at the Naivasha screening facility in Kenya [2,109,114,115]. Due to those tremendous efforts, KS23-5 and KS23-6 have been identified as excellent MLN resistance donors [66]. These two lines were subsequently used to form diverse mapping or breeding populations with the assistance of molecular markers. For instance, CIMMYT has developed and genotyped four F2 populations using diversity arrays technology (DArT) sequencing, and a major QTL has been identified in KS23-6 through a cross between KS23-6/Oh28 and other three different populations (KS23/CZL0005, KS23-5/CML545 and KS23-6/ZLO3018) [66,115]. Furthermore, the two major QTLs from KS23-6 were introgressed into 2400 Backcross 3 (BC3) F2 lines, which were subsequently genotyped using Kompetitive allele specific PCR (KASP) technology under field conditions in Kenya, among which, nineteen BC3F2 lines showing an improved resistance to MLN were screened out [73]. The identified recessively inherited major-effect QTL was further finely mapped and tightly linked flanking markers at Chr 6 in the 157 Mb region were developed [116]. This QTL is widely used to introgress most of the elite lines which are drought-tolerant, high-yielding, low-soil-N-tolerant lines, but they lack the MLN resistance. Until now, more than 145 elite lines were introgressed with an MLN-resistant QTL (qMLN06_157) through marker-assisted back crossing (MABC). These introgressed lines are redistributed into all of the partners to be used in their breeding program. In addition, the identified MLN resistance linked markers are routinely used as part of a forward breeding strategy, where all of the DH lines are subjected to select with these markers before test cross formation and early yield trials are planned in maize breeding in Eastern Africa. In CIMMYT alone, since the identification of these markers, >26,000 DH lines have been screened for MLN resistance, and >72% of the lines were selected, and the remaining lines were discarded based on presence of favorable alleles in the lines (unpublished results from M.G.; it is routine breeding activity).

The DH lines are ideal materials for expediting the breeding process. CIMMYT has evaluated more than 3000 DH lines using GBS technology for MLN resistance screening [7]. In a pre-breeding MLN resistance assessment, the DH lines with a high general combining ability (GCA), including CKDHL120918 and CKDHL0500, were identified [69]. The DH lines, CKDHL120918 and CKDHL120312, were also validated for their tolerance to both SCMV and MCMV prior to developing Kenyan adapted maize varieties with an improved MLN resistance via tolerance introgression [117]. Moreover, three DH populations were subjected to GWAS, joint linkage association mapping (JLAM) and linkage mapping to dissect the genetic basis of MLN and MCMV resistance [67]. Continuous efforts to search for new maize or alternative sources for resistance to facilitate the development of new MLN-resistant/tolerant maize varieties are ongoing, with there being possible positive outcomes in the near future.

Marker-assisted selection is increasingly being used to accelerate the breeding progress upon the successful identification of QTLs and markers linked to MLN resistance [72,87]. For instance, the markers linked to putative MLN resistance QTLs were determined and selected from six donor parent (DP) lines [78,80], and they were subsequently subjected to introgression assessment by targeting several potential loci in a series of RP (Recurrent parent)×DP combinations [69]. Through the evaluation of introgression efficacy, from one to six newly converted versions per RP line were selected from a 25 RP background of the above combinations in the MABC pipeline, and they were advanced and found to show an enhanced mean yield production with improved MLN resistance [7]. Excitingly, 18 hybrids with better MLN resistance levels had been released in East Africa by 2019 [7], demonstrating the success of this strategy in MLN breeding program. Currently for MLN, recessively inherited, major-effect QTLs with tightly linked flanking markers are available, which helps us to select the MLN resistance lines with >97% accuracy. However, these major-effect QTLs are present only in the KS23 line background, so increasing these QTLs in the breeding lines is one of the major priorities. Through marker-assisted selection, several donor sources have been developed and used in DH line development. Several MLN-resistant hybrids have been developed and allocated to private companies and NARS partners for scaling. The major drawback of this QTL is that since it is recessively inherited, the presence of this QTL on all of the parents of the commercial hybrid is necessary, which is tedious for the breeding program to develop a product for diverse markets. The further editing of this specific QTL is in progress in CIMMYT, and gene-edited MLN resistance lines and hybrids are also soon expected to be on the market.

### 6.2. Application of Genomic Selection in MLN Resistance Breeding

The traditional marker-assisted selection (MAS) used for quantitative traits consists of two steps, identifying the QTLs, and estimating their effects. However, the MAS using a single bi-parental population for the QTL mapping of complex polygenic traits, such as MLN resistance, controlled by many loci with a small effect, is inadequate for capturing all of the allelic diversity and complexity as in real breeding program. In addition, the higher labor cost, low efficiency and uncertainty of MAS in exploring the QTLs hinders its application in locus identification and the effect estimation of small-effect QTLs, thereby slowing down crop improvement through the breeding program [118,119]. As an updated form of MAS, genomic selection (GS), also known as GP, is drawing increasing attention from researchers and breeders, benefiting from the drop in the cost of the DNA marker and sequencing processes [120,121].

GS was first proposed by Meuwissen and colleagues to be used in animal breeding, which utilizes the genome-wide markers to predict the allele effects of the breeding value of each individual in untested populations and select the potential individuals for further breeding process [122]. This method employs a strategy based on the linkage disequilibrium (LD) and the association between the markers and phenotypes generated from a training population to construct a prediction model of the phenotypic traits and genome-wide markers. This model is subsequently used to assess the genotypes of test populations, thereby, the genomic estimated breeding value (GEBV) for the individuals in a test population is predicted without phenotypic evaluation. Compared to traditional MAS, GS is more efficient to estimate the effects of all of the loci at a genome-wide level, and it is capable of computing the GEBV of both the major and minor effect genes, therefore achieving much more comprehensive and reliable selection outcomes [123], thus, it is rapidly used in both animal and plant breeding programs [120,121,124], including those for MLN resistance.

To assess the potential of GS for MLN resistance, the ridge-regression BLUP (RR-BLUP) with fivefold cross-validation was used for within and across an AM population [80]. It showed that the population size and marker numbers often affected the prediction accuracy. Overall, the prediction accuracy for the IMAS-AM panel was higher than it was for the DTMA-AM panel, whereas a relatively higher accuracy was found when the prediction was carried out based on random markers together with the MLN-associated markers compared to that for the random markers alone [80]. A similar approach to GS for both MCMV and MLN resistance was also used in an IMAS panel and three DH populations, along with linkage mapping and GWAS [67]. It was found that the average accuracies within the population for the IMAS panel and individual DH populations ranged from 0.21 to 0.78 for MCMV-DS and 0.29 to 0.95 for MCMV-AUDPC, respectively. Moreover, the prediction accuracy for the MLN-DS and MLN-AUDPC ranges from 0.46 to 0.62 and from 0.46 to 0.87, respectively, suggesting that there are overall higher prediction accuracies for MLN than those which are achieved for MCMV in terms of both of the traits. Moreover, the combined DH populations alone displayed a higher accuracy than all of the tested populations combined did [67]. In line with above results, another study showed that the GS accuracy was increased with the marker density and the population size at a constant marker density for both the MLN-DS and AUDPC values [91]. The promising prediction accuracy was also found in a separate GS test using four bi-parental populations, which produced a prediction accuracy of >0.65 across three populations for both MLN-early and MLN-later DS traits, and the prediction accuracy was slightly improved when the MLN resistance-associated markers wereincluded in the model [78]. Similarly, a more recent work obtained 0.50 and 0.58 prediction accuracies for MLN-DS and AUDPC, respectively, in a panel consisting of 879 DH lines derived from 26 bi-parental populations [81]. Therefore, GS shows great potential for accelerating the MLN breeding program by accounting both the major- and minor-effect QTLs in the process, thus, it will significantly shorten the breeding cycle and reduce the overall costs.6.3. Genome-Editing for MLN Resistance Breeding.

In the last decade, significant effort has been made towards developing MLN-resistant/tolerant maize lines via conventional backcross breeding. However, the traditional backcrossing approaches are time-consuming and costly, thus, they greatly retard the breeding process [125]. However, the new tools such as genome-editing technologies can accelerate the breeding program by creating novel breeding lines that are resistant to MLN disease. For instance, CIMMYT has partnered with Corteva agri-science to apply CRISPR-Cas9 to a maize breeding program to address MLN [126,127]. The use of CRISPR-Cas9 could speed up breeding time to 2–3 years, unlike that of 6–10 years for conventional breeding. The spin-offs for a reduced breeding period will be quick access to affordable MLN tolerant maize varieties by resource-limited smallholder farmers (https://mln.cimmyt.org) (accessed on 12 September 2022). CIMMYT and Corteva agri-science are utilizing *Baby Boom-Wuschel2* transformation protocols, and so far, four maize lines that are parents to the commercial hybrids in Africa have been transformed with MLN-resistant alleles from the donor line KS23-6. Genome sequencing played a key role in defining a suitable genome editing strategy for accelerating MLN resistance breeding [128,129]. In the case of MCMV detection, a rapid and efficient platform using a CRISPR/Cas12a-based detection system combined with reverse transcription and recombinase-aided amplification was also developed. This approach utilize the visual nucleic acid detection method, in which Cas12/protein could target and cleave the MCMV coat protein gene, subsequently producing fluorescence signal [130]. Albeit there are very few cases reported so far of the successful application of genome-editing technology in MLN research. Nonetheless, this method has great potential in breeding novel maize varieties with enhanced MLN resistance.

## 7. Conclusions and Perspectives

Due to its devastating nature and rapid spread, MLN is a great threat, not only to food security, but also to the livelihoods in SSA. To curb its spread in ESA and other regions worldwide, a combination of multiple strategies is crucial. These will entail both the traditional disease control approaches and the novel technologies that include genome characterization, genetic dissection and molecular marker-assisted breeding. Thus far, CIMMYT and their partner organizations have deployed various approaches to combat the spread of the disease in the region. These include identifying the resistant germplasms, characterizing the genetic basis of MLN resistance maize genotypes, the identification of the loci associated with resistance, as well as generating novel varieties through conventional MAS breeding programs and DH lines. The tremendous efforts from the research community and the farmers have led to the development of a series of elite MLN-resistant lines with great potential, namely, KS23-5 and KS23-6. These lines have proved to be excellent donors, bearing resistance loci that can be deployed in candidate gene mapping and introgression into commercial lines. The activities by CIMMYT and their partner organizations have led to availability of numerous great germplasm resources, multiple QTLs/SNPs related to MLN disease, among which Chr 3 and Chr 6 seem to bear a series of hot spots, revealing their great potential for the improvement of MLN resistance. Furthermore, the recessively inherited major-effect QTL (qMLN06_157) is very effective in controlling MLN; however, depending on a single major QTL also carries the risk of resistance breakdown. Therefore, further studies on finding new sources of resistance are still in progress.

Despite the rapid research and breeding progress that have been achieved thus far since the first report of MLN disease in EA, there have been no reports of a promising gene with major effects on disease control being cloned. This clearly demonstrates the complex nature of MLN resistance which is controlled by large numbers of minor-effect loci and is putatively influenced by the environmental conditions. Furthermore, the molecular and genetic mechanisms underlying MLN resistance remain unfathomed. Due to third-generation sequencing technologies, novel functional genomics tools, genome-editing and DH technology, the breeding period can largely be shortened, thus, the deployment of novel lines within a short time in East and Southern Africa is feasible.

## Figures and Tables

**Figure 1 viruses-14-02765-f001:**
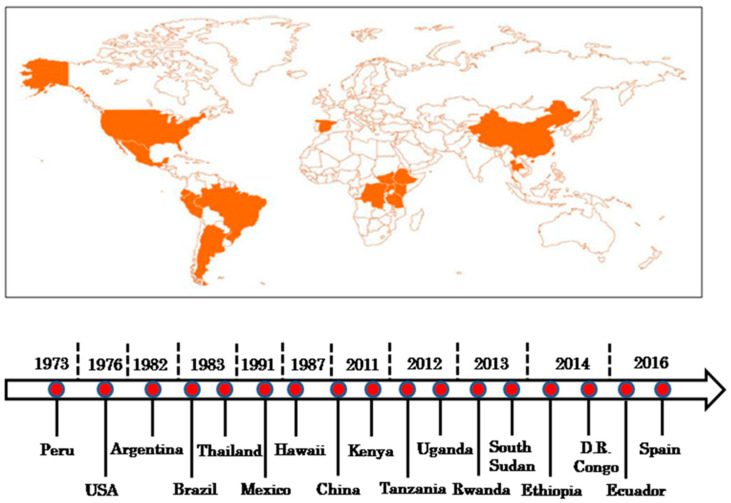
Global occurrence and distribution of maize necrosis disease and historical perspective of MCMV (modified based on photo source: Suresh, L.M.).

**Figure 2 viruses-14-02765-f002:**
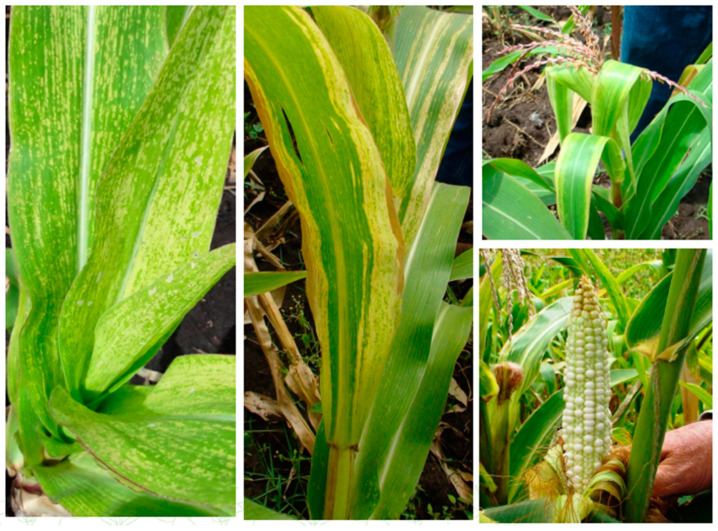
Maize Lethal Necrosis disease symptoms (photo source: Suresh, L.M.).

**Table 1 viruses-14-02765-t001:** Summary of genetic loci identified for MLN disease resistance.

No.	Loci Names (QTLs or SNPs)	Genetic Approaches	Chromosomes	Positions (cM/Mb)	Lines/Populations	Virus	Genetic Effects	References
1	qMLN06_157	Linkage mapping + association mapping	Chr 6	/155.6 /156.5	F2 population: SG1 (KS23-5 × CZL00025), SG2 (KS23-5 × CML545), SG3 (KS23-6 × CML539), SG4 (CML494 × CZL068), SG5 (DTP-F46 × CML442).	MCMV and SCMV	55–70%	[65]
2		Joint mapping	Chr 1, 2, 3, 5, 6, 10		F2 populations: (KS23-5, KS23-6, N211, DR, and Oh1VI) × Oh28 (S) RIL: Oh1VI × Oh28 (S)	MCMV	9~78%	[66]
3 4 5 6	qMCMV3-108 qMLN3-108 qMCMV6-17 qMLN6-17	GWAS+Linkage mapping + GP	Chr 3 Chr 3 Chr 6 Chr 6	39/ 39/ 38/ 38/	380 lines of Association Mapping Panel; Three DH populations	MCMV and MLN	49.87% (DS) + 58.70% (AUDPC) 21.43% (DS) + 23.73% (AUDPC) 29.4% 17.6% (DS) and 22.9% (AUDPC)	[67]
7		GWAS Linkage mapping	Chr 3, 4, 5, 8 Chr 10		282 lines of Association Mapping Panel; RIL: B73 (S) × CML333 (R).	MCMV	6.6–15.1% each SNP 16.3% for qMCMV	[68]
8 9 10 11 12 13 14	qMLN3_130 qMLN3_142 qMLN5_190 qMLN5_202 qMLN6_85 qMLN6_157 qMLN9_142	Linkage mapping	Chr 3 Chr 3 Chr 5 Chr 5 Chr 6 Chr 6 Chr 9.	/125.1 /52.8;/68.5;/133.0 /190.6 /200.9 /85.2; /156.3 /132.7	306 F3 lines: pop1:CKDHL120918 × CML494, pop2: CML543 × CML494, pop3:CKDHL120918 × CML543, pop4:CKLTI0227 × CKDHL120918, pop5:CKDHL0089×CKDHL120918, pop6:CKDHL0221 × CKDHL120312, pop7:CKDHL0089 × CML494.	SCMV and MCMV		[72]
15	16 main effect QTL for MLN-early and 10 for MLN-late	Joint linkage association + Genome-wide prediction mapping	Chr 3 Chr 6 Chr 9		F3 lines: Pop 1: 229 (CML543 × LaPostaSeqC7-F71-1-2-1-2-B-B-B-B) pop 2: 200 (CML543 × CML444) pop 3: 260 (CML444 × CML539) pop 4: 124 (Mo37 × CML144)		3.9~43.8%	[78]
16 17 18	*Scm2a*, *Scm2b*, *Scm1.*	Linkage mapping	Chr 3 Chr 3 Chr6	23.5/ 46.6/ 27.5/	150 F2:3 lines from L520 (R) and L19 (S)	SCMV	13.3%, 41.9% and 7.7% of PVE, respectively	[79]
19		GWAS	Across 10 Chrs		IMAS-AM: 380 lines; DTMA-AM: 235 lines.	MLND	8~10% and 14–18% each, respectively.	[80]
20	12 SNPs for disease severity; 10 SNPs for AUDPC	GWAS + GP	Chr 3, 4,6~10 (DS) Chr 1,2,4,6~10 (AUDPC)		879 DH lines	MLND		[81]
21	*Scmv1*	Linkage mapping Association mapping	Chr 6		177 F2 lines: F7^RR/RR^ × F7^RR/SS^ RILs: Zheng58(S) × Chang7-2(R), X178 (R) × HuangC(S)	SCMV		[82]
25 26	*Scmv1*, *Scmv2*.	Linkage mapping Association mapping	Chr 6 Chr 3	30.7/ 18.4/	121 F3 lines: F7 (S) ×FAP1360A (R)	SCMV	15~62%	[83,84,85,86]

**Table 2 viruses-14-02765-t002:** Candidate genes identified for MLN disease resistance.

No.	Name of Candidate Gene	Gene ID or Tag	Viruses Involved	References
1	*ZmTrx-h*	Zm00001d035390	SCMV	[82,85,86]
2	*ZmABP1*	Zm00001d041711	SCMV	[83,84,85]
3	*metallothionein-like protein*	605018B04.x1 (UP|Q5U7K6_9POAL (Q5U7K6))	SCMV, MDMV	[101]
4	*S-adenosylmethionine synthetase*	946126A02.y1 (UP|METK_ORYSA (P46611))	SCMV, MDMV	[101]
5	*germin-like protein 4*	za72g09.b50 (UP|O49000_ORYSA (O49000))	SCMV, MDMV	[101]
6	*large subunit 26S ribosomal RNA*	605018B03.x1 (gb|AF036494.1|AF036494)	SCMV, MDMV	[101]
7	*14-3-3- like protein GF14-6*	Zm06_09h07_R (UP|14331_MAIZE (P49106))	SCMV, MDMV	[101]
8	*ZmPDIL-1*	GRMZM2G091481_P01	MCMV, SCMV (susceptible)	[96,97]
9	*ZmPrx5*	GRMZM2G036921_P01	MCMV	[97]
10	*ZmElc*	KJ811537/Zm00001d037277	SCMV (susceptible), MCMV (resistant)	[94]
11	*SceIF4Ea/SceIF4Eb/* *SceIF(iso)4E*	MW547070/MW547071/MW547072	MCMV (susceptible)	[94,95]
12	*ScnCBP*	KX757019	MCMV (susceptible)	[94,95]
13	*ZmTGL*	NP 001149280	SCMV	[103]
14	*ZmPAL*	Zm00001d017274	SCMV	[104]
15	*ZmDWF4*	Zm00001d028325	MCMV	[105]
16	*ZmNR*	Zm00001d049995	MCMV	[105]
17	*ZmARF3/30*	Zm00001d001879/Zm00001d026590	MCMV (susceptible)	[98]
18	*ZmPAO1*	Zm00001d024281	MCMV (susceptible)	[98]
19	*ZmCAT1/2/3*	NM_001254879.2/NM_001111840.2/NM_001363892.1	MCMV	[30]

## Data Availability

Not applicable.

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
