# Peer review of "Fighting Death for Living: Recent Advances in Molecular and Genetic Mechanisms Underlying Maize Lethal Necrosis Disease Resistance"

_viruses, 2022, doi:10.3390/v14122765_

Round 1

Reviewer 1 Report

This review summerizes the recent advantages in MLN studies on the way to find efficient resistance. The manuscript is well written and I only have minor suggestions:

line 89:non-potyviruses are (P lower case)
line 92: exchange "unlike MCMV alone" for "in comparison to MCMV single infected plants is due to...."
line 95: Machlomovirus (one word), family Tombusviridae (italics)
line 98: range, delete s
lines 104/9: readthrough in one word
line 110: what is expressed from viral vector? thee p31? systemic infection?
line 115: has
line 137: R: padi - use full name
line 141: have largely been ...(delete previous "been")
line 146: which polero- and totiviruses?
line 149: MYMV family and/or genus?
line 296: of ONLY A limited number ...
line 300: Delete THE at the begining of sentence
line 310: infection are
line 313: and was
line 319: high virulence against host factots?
line 572:promosing geneS with major effects ON DISEASE CONTROL cloned. This clearly deminstrates the...

Author Response

Responses to  Reviewers’ comments

Review 1:

 This review summerizes the recent advantages in MLN studies on the way to find efficient resistance. The manuscript is well written and I only have minor suggestions:
 line 89:non-potyviruses are (P lower case)

A: thanks for your comments! “Potyviruses” was changed to “potyviruses”.
line 92: exchange "unlike MCMV alone" for "in comparison to MCMV single infected plants is due to...."

A: we revised the sentence to “in comparison to that infected by MCMV alone…”
line 95: Machlomovirus (one word), family Tombusviridae (italics)

A: we changed “Machlomovirus” to oneword, added “family” in front of Tombusviridae and italicized “Tombusviridae”.
line 98: range, delete s

A: the “s” was deleted.
lines 104/9: readthrough in one word

A: we made “readthrough” one word on both lines.
line 110: what is expressed from viral vector? thee p31? systemic infection?

A: we changed it to “the expression of p31”.
line 115: has

A: we changed “have” to “has”.
line 137: R: padi - use full name

A: we supplemented the full name “RhopalosiphumpadiL.”
line 141: have largely been ...(delete previous "been")

A: thanks! The extra “been” was deleted.
line 146: which polero- and totiviruses?

A: we are sorry for that we cited a wrong reference. We replaced the old reference with new one (#55, Wamaita et al., 2018), deleted “totiviruses”and stated that the polerovirus includes Maize Yellow Dwarf Virus (MYDV)-RMV and Maize Yellow Mosaic Virus (MYMV), and Barley Virus G (BVG).
line 149: MYMV family and/or genus?

A: we stated that MYMV belongs to polerovirus genus.
line 296: of ONLY A limited number ...

A: “only a” was added.
line 300: Delete THE at the begining of sentence

A: “The” at the beginning of sentence was deleted.
line 310: infection are

A: to avoid the misunderstanding, we modified the description to “the roles of host factors for resistanceto MCMV and/or SCMV are…”
line 313: and was

A: we changed “was” to “is”.
line 319: high virulence against host factots?

A: we changed it to “high virulence on host”.
line 572:promosinggeneS with major effects ON DISEASE CONTROL cloned. This clearly deminstrates the...

A: thanks a lot for your comments! We added “on disease control” and changed “demonstrated” to “demonstrates”.

Reviewer 2 Report

·       Add a separate figure about maize lethal necrosis disease symptoms

·       Add a separate figure about the historical perspective of maize lethal necrosis disease

·       Add a separate figure about maize lethal necrosis disease distribution around the world

·       Align table 1 properly, provide the genetic (cM) and physical distance (kp) for genetic loci identified for MLND resistance

·       Write a separate paragraph about vector details of MLND resistance

·       Write a separate paragraph about the genetics of MLND resistance (Complex genetic nature) before section 5.

·       Prepare a separate table for candidate genes identified for MLND resistance

Author Response

Responses to Reviewer 2:

Add a separate figure about maize lethal necrosis disease symptoms

A: a separate figure (Figure 2) was provided for MLN disease symptoms. In addition, we described more details about MLN disease symptoms.

Add a separate figure about the historical perspective of maize lethal necrosis disease

A: a separate figure (Figure 1) was provided for the historical perspective of maize lethal necrosis disease.

Add a separate figure about maize lethal necrosis disease distribution around the world

A: a separate figure (Figure 1) was provided for MLN disease distribution.

Align table 1 properly, provide the genetic (cM) and physical distance (kp) for genetic loci identified for MLND resistance

A: thanks for your constructive suggestions. We aligned the contents in Table 1 and reformat the table so that the items can be easily separated and recognized. Also, wedeleted the column for “application status” that is overlapped with the new Table 2.Besides, we tried our best tosupplement genetic (cM) and physical distancesfor only someof the loci, if reported in the literature; however, we are not capable of providing the information for those not reported in the publication.

Write a separate paragraph about vector details of MLND resistance

A: we expandedthe description on MLN vectors by providing more detailsfor MCMV and SCMV, respectively (see the end paragraph of 3.1 and 3.2). Based on the contents supplemented, we replaced the old citation #54 with a more appropriate reference.

Write a separate paragraph about the genetics of MLND resistance (Complex genetic nature) before section 5.

A: According to your suggestion, we wrote several sentences to describe the complex genetic nature for plant disease resistance in the before section 5.1. In brief “The genetics of MLN is a combination of both simple and complex, as it is controlled by the combination of genetic resistance to SCMV, MCMV as well as their interactions. Earlier studies clearly showed that SCMV is controlled by two major effect genomic region on chromosome 3 and 6 and many small effect QTL distributed on different chromosomes. On the other hand, MCMV had major effect resistance gene on chromosome 6 (Jones et al., 2019; Murithi et al., 2021) and many minor effect QTL distributed on chromosome 3, 4, 5 and 8 (Sitonik et al., 2019; Ohlson et al., 2022). Genetic studies also support both SCMV and MCMV and MLN resistance are governed mainly by additive effects and to some extent with non-additive effects (Beyene et al., 2017; Nyaga et al., 2020). This suggests MLN resistance in controlled by few major effect genes and possible to improve by rapid recurrent selection.”

 Prepare a separate table for candidate genes identified for MLND resistance.

A: we provided an additional table (Table 2) for candidate genes identified for MLN resistance.

Round 2

Reviewer 2 Report

Accept the manuscript in present form.

Author Response

thank you very much!